# Bacterial Diversity and Antibiotic Resistance in Patients with Diabetic Foot Osteomyelitis

**DOI:** 10.3390/antibiotics12020212

**Published:** 2023-01-19

**Authors:** Francisco Javier Álvaro-Afonso, Yolanda García-Álvarez, Aroa Tardáguila-García, Marta García-Madrid, Mateo López-Moral, José Luis Lázaro-Martínez

**Affiliations:** 1Diabetic Foot Unit, Clínica Universitaria de Podología, Facultad de Enfermería, Fisioterapia y Podología, Universidad Complutense de Madrid, 28040 Madrid, Spain; 2Instituto de Investigación Sanitaria del Hospital Clínico San Carlos (IdISSC), 28040 Madrid, Spain

**Keywords:** diabetic foot, diabetic foot ulcers, diabetic foot infection, diabetic foot osteomyelitis, microbiology, antibiotic resistance

## Abstract

This study analysed the bacterial diversity, antibiotic susceptibility, and resistance in patients with complications of diabetic foot osteomyelitis (DFO). A retrospective observational study was carried out between September 2019 and September 2022 and involved 215 outpatients with a diagnosis of DFO at a specialized diabetic foot unit. A total of 204 positive bone cultures were isolated, including 62.7% monomicrobial cultures, and 37.3% were formed with at least two microorganisms. We observed that *Proteus* spp., Coagulase-negative staphylococci (CoNS), *Staphylococcus aureus*, *Pseudomonas aeruginosa*, *Escherichia coli*, and *Corynebacterium* were the most frequently isolated microorganisms and accounted for more than 10% of the DFO cases. With stratification by Gram-positive (GP) and Gram-negative (GN) bacteria, we observed that 91.6% of cultures presented at least one GP bacteria species, and 50.4% presented at least one GN bacteria species. The most common GP species were CoNS (29%), *S. aureus* (25.8%), and *Corynebacterium* spp. (14%). The most frequent GN species consisted of *Proteus* spp. (32%), *P. aeruginosa* (23.3%), and *E. coli* (17.5%). The main antibiotics with resistance to GP-dominated infections were penicillins without β-lactamase inhibitor, and those in GN-dominated infections were sulfonamides and penicillins without β-lactamase. Significant differences were not observed in mean healing time in DFU with acute osteomyelitis (12.76 weeks (4.50;18)) compared to chronic osteomyelitis (15.31 weeks (7;18.25); *p* = 0.101) and when comparing cases with soft tissue infection (15.95 (6;20)) and those without such an infection (16.59 (7.25;19.75), *p* = 0.618). This study shows that when treatment of DFO is based on early surgical treatment, the type of DFO and the presence of soft infection are not associated with different or worse prognoses.

## 1. Introduction

Diabetic foot ulcers (DFUs) are one of the most frequent, serious, and costly complications associated with diabetes mellitus (DM). People with diabetes are likely to develop a DFU in approximately 19 to 34% of cases during their lifetime [1], and around 50% of them will become infected [2]. Diabetic foot infection (DFI) might be related to more severe outcomes that contribute to morbidity, increasing costs, and decreased quality of life [3,4].

These infections frequently cause osteomyelitis, spreading contiguously to deep tissues if they reach bone tissue [5]. Diabetic foot osteomyelitis (DFO) occurs in up to 20% and 50–60% of patients with moderate and severe infections, respectively [6,7,8]. Some studies have tried to demonstrate the risk factors for the development of DFI, which include some local characteristics such as DFU mean evolution time > 30 days, traumatic aetiology [9], a wound that has extended to bone [10], and others such as the presence of history of a previously wound, recurrent wounds, a previous amputation [11], peripheral arterial disease (PAD) [10], loss of protective sensation [11], and the presence of renal failure [12].

Antibiotic treatment is a key part of DFI management [13], and an appropriate antibiotic should be initiated promptly. Accordingly, the Infectious Disease Society of America (IDSA) provides guidelines for the choice of empirical antibiotics [5]. Clinicians should follow these guidelines before the result of a bacteriological culture becomes available [14,15].

Despite these recommendations, multidrug resistance poses a serious problem and is a major public health threat [16]. The main risk factors for multidrug resistance are the overuse and inappropriate use of antibiotics, previous antibiotic therapy, previous amputation [17], frequent hospital admission, and the chronic course of a wound [18]. Pathogenic organisms are heterogeneous and can vary depending on several factors, such as geographical area, socioeconomic status [15], and specific wound and patient characteristics, such as the mean duration of the ulcer, depth, or peripheral arterial disease (PAD) [14,19]. The pathogen distribution usually follows a trend that could help clinicians to choose empirical antibiotic regimens [20]. *Staphylococcus aureus*, *Streptococcus,* and *Enterococcus* are the predominant Gram-positive (GP) pathogens, in addition to *Enterobacteriaceae* and *Pseudomonas aeruginosa*, which are well recognized as the predominant Gram-negative (GN) pathogens in DFI [21].

Thus, DFO antibiotic management remains a challenge for clinicians. Due to the limitations of obtaining bone culture, many patients are treated with empirical antibiotics and are exposed to the drugs for a long time. To the best of our knowledge, few studies have described the specific microbial profile of DFO. A better understanding of the behaviour of bacteria in DFO could help clinicians to select the correct antibiotic therapy and predict the prognosis to reduce treatment failure, antimicrobial resistance, adverse events, and costs. Therefore, we mainly aimed to describe the bacterial diversity and the antibiotic susceptibility and resistance in patients with DFUs with complications of DFO. We also analysed the influence of the presence of soft tissue infection and the type of DFO on ulcer healing times.

## 2. Materials and Methods

### 2.1. Study Design and Participants

Between September 2019 and September 2022, a retrospective observational study was carried out at a specialised Diabetic Foot Unit of the Complutense University of Madrid, where the patients received treatment for the infection. The study involved 215 patients with DM with a diagnosis of osteo-myelitis. This diabetic foot unit is a multidisciplinary centre, coordinated by a podiatrist linked with the infectious disease and vascular surgery departments, specialising in the management of patients with diabetic foot complications.

The criteria for selection were as follows: people with DM > 18 years of age; the presence of ulcer diagnosed with DFO (clinically); DFU treated by surgical intervention; and the presence of at least grade 3 “O” DFUs according to the PEDIS classification [22] and Texas University Wound Classification grades IIIB and IIID [23]. We excluded patients with a diagnosis of critical limb ischemia. [24]. Moreover, patients whose samples were insufficient for microbiological laboratory testing were excluded.

### 2.2. Clinical Diagnosis of DFO

The diagnosis of DFO was made clinically by the combination of regular X-rays and the probe-to-bone (PTB) test. This combination has shown high sensitivity, specificity, and reproducibility in the diagnosis of DFO [25,26]. The PTB test was performed with sterile metal forceps, such as Halsted mosquito forceps, and was considered positive when the investigator could feel a sandy or hard surface. Two standard views of the X-rays were obtained, and the result was considered positive for DFO if it showed cortical disruption, periosteal elevation, a sequestrum or involucrum, or gross bone destruction. Both tests had to be considered positive [27].

### 2.3. DFU Assessment

Diabetic neuropathy was diagnosed in cases where patients felt nothing during one of the following tests: Semmes-Weinstein monofilament 5.07/10 g (Novalab Ibérica, Alcalá de Henares, Madrid, Spain) and Horwell biotensiometer (Me.Te.Da. S.r.l., San Benedetto del Tronto, Italy). [28]. Vascular screening was carried out by distal pulse palpation, ankle brachial index (ABI), toe brachial index (TBI) and transcutaneous oxygen pressure (TcPO_2_). Peripheral arterial disease (PAD) was diagnosed if the patient showed an absence of distal pulses (posterior tibial pulse and dorsalis pedis), ABI < 0.9 (or in cases of arterial calcification (ABI > 1.4), TBI < 0.7, and TcPO_2_ < 30 mmHg [24,29,30].

Soft tissue infection was considered when local classic clinical signs of infection were shown through pain, flushing, warmth, and/or swelling [5]. DFO was managed by surgical treatment [13] according to previously published recommendations [31]. The same surgeon (J.L.L.-M.) performed all surgical procedures using “conservative” surgery to remove necrotic bone and soft tissue without amputation of any part of the foot [32]. The surgeon is a specialist in diabetic foot surgery with more than 20 years of experience. Post-surgical antibiotics were maintained in all patients for at least one week. The antibiotic was adjusted with the antibiogram results if the bone culture was positive [33] and was empirical if the culture was negative [5].

### 2.4. Microbiological Analysis of Bone Samples

After discontinuation of antibiotic treatment for 48–72 h, bone samples were taken from all patients using the aseptic technique [22] The ulcer was debrided, and povidone-iodine was applied to the perilesional skin and washed with saline, avoiding contamination [33]. Intraoperatively, the bone sample (if large enough) was split to send a portion for microbiological analysis and another portion of bone for histopathological analysis [22]. The protocol established by our laboratory was used for sample collection and transport [34].

The samples were mechanically homogenised by the same microbiologist in 1 mL of sterile phosphate-buffered saline at pH 7.4 (PBS, Sigma Aldrich, St Louis, MO, USA) for 5 min. Subsequently, the homogenised samples were plated on a culture medium and incubated at 35 °C for 48 h. Isolated microorganisms were identified by convention-al Gram staining methods [35] and classified as mono- or polymicrobial [36]. Susceptibility testing was performed in accordance with Clinical and Laboratory Standards by the disk diffusion method [37].

### 2.5. Histopathological Analysis of Bone Samples

Bone sample processing for pathology analyses was done as follows. The same pathologist evaluated the bone samples, which were sent in 10% buffered formalin for 24–48 h. The sample was then decalcified, carved, embedded in paraffin, cut with the microtome, stained with haematoxylin-eosin, observed under the microscope [38], and classified as acute or chronic DFO [39]. Acute DFO: multifocal infiltrate in the bone marrow of plasma cells, polymorphonuclear neutrophils and lymphocytes with clear polymorphonuclear predominance, and a variable proportion of concentrations of necrosis. Chronic DFO: multifocal infiltrate in the bone marrow of plasma cells and lymphocytes with mononuclear predominance and some concentrations of reshaped bone with a variable fibrosis or osteoid formation [39].

### 2.6. Statistical Analysis

SPSS^®^ version 20.0 for iOS (SPSS, Inc., Chicago, IL, USA) was used. Descriptive analyses were performed to process the data. We calculated the means and standard deviations for quantitative variables. We calculated the frequency distributions and percentages for qualitative variables. For quantitative variables, the assumption of normality was verified using the Kolmogorov-Smirnov test. A Student’s *t*-test was performed for quantitative variables distributed normally, and the Mann-Whitney U test was used for abnormally distributed quantitative parameters.

To identify differences in qualitative variables the chi-square test was used. To describe survival time to healing, the Kaplan-Meier method was used, and to compare this survival between the acute and chronic DFO groups the log-rank test was used. *p*-values < 0.05 were considered significant with a 95% confidence interval.

### 2.7. Ethical Considerations

Ethics committee approval for the study was obtained from the local Ethics Committee of the Hospital Clínico San Carlos, Madrid, Spain (20/092). The study was conducted in accordance with the ethical code of the Declaration of Helsinki [40]. Informed consent of the patients was not required due to the retrospective nature of the study.

## 3. Results

### 3.1. Study Population

In the analysis period, we identified 215 patient cases, which were 82.3% (n = 177) men. The mean age was 64.71 ± 11.1 years. There were 190 patients (88.4%) who had type 2 diabetes, whereas the remainder had type 1 diabetes. The mean time of evolution of diabetes was 16.99 ± 10.99 years, and the mean haemoglobin A1c was 7.6 ± 1.46% (60 ± 11.5 mmol/mol) [41]. Table 1 shows the clinical characteristics of our study population.

All subjects (100%) had sensory neuropathy. There were 13 (6.4%) cases of neuroischemic DFUs. The mean ankle–brachial index was 0.97 ± 0.54, indicating no significant problems with arterial perfusion and that these ulcers were primarily neuropathic. The mean wound tissue oxygen level was 34.8 ± 13.12 mmHg. A total of 95.3% (n = 205) of the wounds pertained to the toes and plantar sides of metatarsal head bones and the sole, 3.7% (n = 8) were in the midfoot, and 0.9% (n = 2) were in the hindfoot. All wounds were categorized according to the Texas classification as grade IIIB (93.6%; n = 202) or IIID (75.3%; n = 13). Soft tissue infection was observed in 73.2% (n = 157) of the wounds, and the mean ulcer duration was 19.41 weeks (3;24). The mean healing time in our study population was 16.19 weeks (6;20).

### 3.2. Bacterial Diversity Isolated in Patients with DFO

A total of 204 positive bone cultures were isolated (94.9%), including 128 (62.7%) monomicrobial cultures and 76 (37.3%) that had formed with at least two microorganisms. Table 2 shows the frequency of the microorganisms isolated in our study population.

### 3.3. Antibiotic Resistance in Patients with DFO

Table 3 and Table 4 describe the analysis results of common GP and GN bacterial resistance to sensitive antibiotics. Table 5 describes the oral antibiotics prescribed in our study population. Postoperative antibiotics were given for a median of 9.5 days (IQR 7–12)

### 3.4. Results of Bone Histopathological Study

We identified 111 patients with pathology results of the resected bone, of which 98 were positive (88.2%). There were 25 (25.5%) patients with acute osteomyelitis and 73 (74.5%) patients with chronic osteomyelitis. The mean times of evolution of the ulcer in patients with acute osteomyelitis versus chronic osteomyelitis were 15.28 (1.5;24) and 25.24 (4.5;28) weeks, respectively (*p* = 0.085).

Significant differences were not observed in the mean healing time in DFU with acute osteomyelitis (12.76 weeks (4.50;18)) compared to chronic osteomyelitis (15.31 weeks (7;18.25); *p* = 0.101). Figure 1 shows the Kaplan-Meir survival curve of the time to wound healing among patients with acute osteomyelitis versus chronic osteomyelitis. There were 141 (65.6%) patients who had soft tissue infection, but compared with patients without soft tissue infection, significant differences were not observed in mean healing time (15.95 (6;20) vs. 16.59 (7.25;19.75) weeks; *p* = 0.618). 

## 4. Discussion

The study sample included 62.7% monomicrobial bone cultures, and 37.3% were formed with at least two microorganisms. These results are in accordance with a recent review [15], where the prevalence of polymicrobial infection in patients with DFI was 22.8%. We observed that *Proteus* spp., CoNS, *S. aureus*, *Pseudomonas aeruginosa*, *E. coli,* and *Corynebacterium* spp. were the most frequently isolated microorganisms, which accounted for more than 10% of our cases with DFO (Table 2). After stratifying by GP and GN bacteria, we observed that 91.6% (108) of cultures presented at least one GP bacteria, and 50.4% (103) presented at least one GN bacteria. The most common GP species were CoNS (29%), *S. aureus* (25.8%), *Corynebacterium* spp. (14%), *Streptococcus* spp. (9.6%), MRSA (8.6%), and *Enterococcus* spp. (6.5%). The most frequent GN species consisted of *Proteus* spp. (32%), *P. aeruginosa* (23.3%), *E. coli* (17.5%), and *Enterobacter* spp. 

To the best of our knowledge, there are currently few studies on the diversity of microorganisms in patients with DFO. Our results were in line with a study by Schmidt et al., who found that the most common microorganisms were MSSA, *S. agalactiae*, *E. faecalis*, *E. faecium*, and CoNS in an observational cohort of 223 patients with DFO [42]. Similar results were found in a study by Johani et al. [43] in 20 consecutive subjects with suspected DFO. They identified that *Corynebacterium* sp. was the most commonly identified microorganism, followed by *Finegoldia* sp., *Staphylococcus* sp., *Streptococcus* sp., *Porphyromonas* sp., and *Anaerococcus* sp. In a study by Zou et al. [44], *Proteus vulgaris* was positively correlated with the infection index in patients with DFO. In a study by Thurler Palomo et al. [45], the most prevalent GN bacterium was *P. aeruginosa* in in-patients with DFI.

According to the antibiotic resistance in our study population, we observed that the resistance rates of GP bacteria to tetracycline and piperacillin-tazobactam were lower (<10%), and the resistance rates to penicillin, cloxacillin, amoxicillin-clavulanic, imipenem, cephalosporins, quinolones, erythromycin, clindamycin and cotrimoxazole, were higher (>10%) (Table 3). In this regard, Dörr et al. [14] suggest that penicillins with β-lactamase inhibitors are the main antibiotics with resistance in cases of GP-dominated infection in a study population of 353 individuals with DFI. 

In our study, the main antibiotics with resistance among GP-dominated infection were penicillins without β-lactamase inhibitors. Nonetheless, we found that 33.3% of all isolated MSSA strains had resistance to penicillin without an inhibitor of β-lactamase (β-LI). In a study by Dörr et al., the rate was almost double (66.7%). The main difference from our study population was that recorded in-patients had infected DFU without specification of how many had DFO. 

In our study population, GN bacteria were less resistant to imipenen, piperacillin/tazobactam, cephalosporins, quinolones, gentamicin, and aztreonam, while they were more resistant to ampicillin and cotrimoxazole (Table 4). Different results were found in a study by Dörr et al. [14], who identified resistance against piperacillin/tazobactam or rather carbapenems with equal efficacy when GN species were present. These differences could be explained by previously mentioned study differences. Nonetheless, a recent systematic review concluded that there was not enough evidence that multidrug-resistant organisms hindered the healing of DFUs [46].

The most frequently prescribed oral antibiotic in our study population was levofloxacin (44.2%), followed by amoxicillin/clavulanate (30.2%) and trimethoprim/sulfamethoxazole (15.8%). Postoperative antibiotics were given for a median of 9.5 days (IQR 7.12). In a recent study by Aragón-Sánchez et al., postoperative antibiotics were given for a median of 14.5 days (IQR 19.8). The main difference was that they treated in-patients with severe DFO [47]. In this regard, another study demonstrated good adherence to oral antibiotic medication in an outpatient clinical setting, independently of the type of infection in patients with DFI [48].

We identified 111 patients with pathology results of the resected bone, of which 98 were positive (88.2%), 25 (25.5%) had acute osteomyelitis, and 73 (74.5%) had chronic osteomyelitis. Significant differences were not observed in mean healing time in DFU with acute osteomyelitis (12.76 weeks (4.50;18)) compared to chronic osteomyelitis (15.31 weeks (7;18.25)); *p* = 0.101. In a previous study, Cecilia-Matilla et al. did not find statistically significant differences when comparing healing times between patients with acute osteomyelitis and chronic osteomyelitis (12 ± 9.7 versus 11.4 ± 10.1 weeks, respectively, *p* = 0.26) [39]. Significant differences were not observed in mean healing time when comparing patients with or without soft tissue infection (15.95 (6;20)) vs. [16.59 (7.25;19.75), *p* = 0.618. These results suggest that when treatment of DFO is based on early surgical treatment, the type of DFO or the presence of soft infection is not associated with different or worse prognoses.

This study had certain limitations. First, we used a retrospective cohort design that included data collection from electronic medical records created for patient care, not for research, and it may contain errors. Second, our data were collected from a single-centre Spanish population and outpatients with DFO who were treated surgically, so the results may not be generalizable to populations with different demographic characteristics or hospitalized patients. Third, we did not analyse some parameters in our study population, such as prior antibiotic use or prior hospitalization.

## 5. Conclusions

In our study population, the majority of bone cultures presented at least GP bacteria, and more than half presented at least GN bacteria. Our results indicated a higher prevalence of CoNS, *S. aureus, Corynebacterium* spp., *Proteus* spp., *P. aeruginosa*, and *E. coli* in bone cultures. The main antibiotics with resistance in GP-dominated infection were penicillins without β-lactamase inhibitors, and those in GN-dominated infection were sulfonamides and penicillins without β-lactamase. This study demonstrates that when treatment of DFO is based on early surgical treatment, the type of DFO or the presence of soft infection is not associated with different or worse prognoses.

## Figures and Tables

**Figure 1 antibiotics-12-00212-f001:**
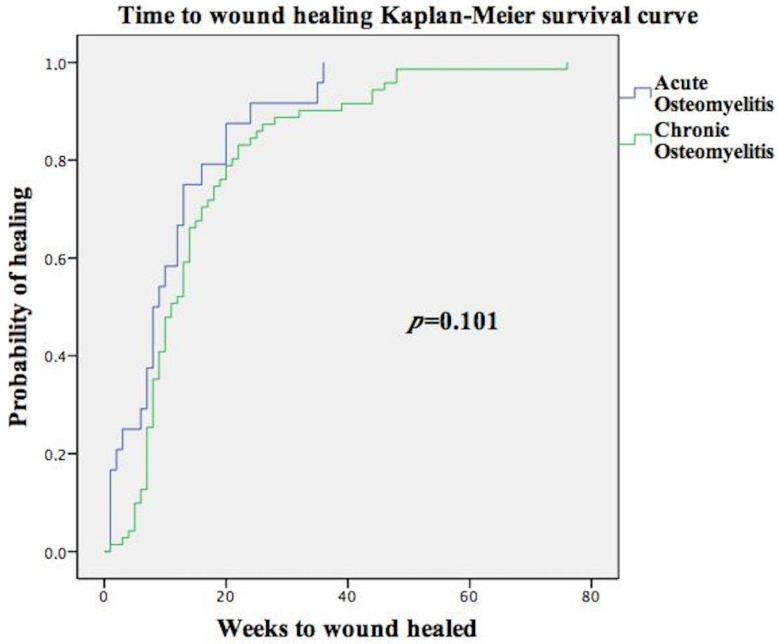
Kaplan-Meir survival curve of time to wound healing among patients with acute osteomyelitis versus chronic osteomyelitis.

**Table 1 antibiotics-12-00212-t001:** Clinical characteristics of the study population. Data are shown as n (%).

Variables	Frequency (n)	Percentage (%)
Hypertension	152	70.8
Hypercholesterolemia	131	60.6
Retinopathy	69	31.7
Nephropathy	49	22.8
Cardiovascular history	71	33.2
Current smoking Former smoker	49 9	23.4 4.3
Previous ulceration	150	70
Previous amputation	131	60.6

**Table 2 antibiotics-12-00212-t002:** Frequency of microorganisms isolated from the DFUs.

Organisms	Number of Pathogens (n)	Percentage (%)
Gram-positive bacteria (n = 186)		
*Staphylococcus aureus* (MSSA)	48	25.8
*Staphylococcus aureus* (MRSA)	16	8.6
*Coagulase-negative Staphylococci* (CoNS)	54	29
*Streptococcus* spp.	18	9.6
*Enterococcus* spp.	12	6.5
*Corynebacterium* spp.	26	14
Gram-negative bacteria (n = 103)		
*Escherichia coli*	18	17.5
*Pseudomonas aeruginosa*	24	23.3
*Klebsiella pneumoniae*	4	3.9
*Proteus* spp.	33	32
*Enterobacter* spp.	10	9.7
*Morganella morgani*	5	4.8
*Klebsiella oxytoca*	3	2.9
*Serratia* spp.	1	0.9
*Citrobacter Koseri*	1	0.9
*Providencia rettgeri*	1	0.9
*Klebsiella pneumoniae*	4	3.9
*Acinetobacter* spp.	1	0.9
*Stenotrophomona maltophilia*	1	0.9
Fungi	4	2.0

Abbreviation: MRSA: methicillin-resistant *Staphylococcus aureus.* MSSA: Methicillin-susceptible *Staphylococcus aureus.*

**Table 3 antibiotics-12-00212-t003:** Resistance analysis of common GP bacteria to sensitive antibiotics.

	GP Bacteria (n = 186)		
Antibiotic	CoNS (n = 54)	MSSA (n = 48)	*Corynebacterium* spp. (n = 26)
Penicillin	25 (46.3%)	16 (33.3%)	13 (50%)
Cloxacillin	14 (25.9%)	0	5 (19.2%)
Ampicillin	-	-	-
Piperacillin	-	-	-
Amoxicillin-clavulanic	14 (25.9%)	0	6 (23.1%)
Piperacillin-tazobactam	-	-	-
Imipenen	14 (25.9%)	0	0
Cefalosporins (Cefuroxime)	14 (25.9%)	0	5 (19.2%)
Quinolones (Ciprofloxacin or Levofloxacin)	18 (33.3%)	4 (8.3%)	12 (46.2%)
Erythromycin	15 (27.8%)	10 (20.8%)	12 (46.2%)
Vancomicin	-	-	-
Clindamicin	10 (18.5%)	5 (10.4%)	12 (46.2%)
Cotrimoxazole	10 (18.5%)	1 (2.1%)	16 (61.5%)
Tetracycline	6 (11.1%)	2 (4.2%)	2 (7.7%)
Linezolid	0	0	0

Abbreviation: GP: Gram positive, MSSA: Methicillin-susceptible *Staphylococcus aureus,* CoNS: Coagulase negative staphylococci. “-”: Not tested.

**Table 4 antibiotics-12-00212-t004:** Resistance analysis of common GN bacteria to sensitive antibiotics.

	GN Bacteria (n = 103)		
Antibiotic	*Proteus* spp. (n = 32)	*Pseudomonas aeruginosa* (n = 24)	*Escherichia coli* (n = 18)
Penicillin	-	-	-
Ampicillin	7 (21.9%)	-	10 (55.6%)
Piperacillin	-	-	-
Piperacillin-tazobactam	-	2 (8.3%)	-
Imipenen	0	0	0
Cefalosporins Cefuroxime	4 (12.5%)	-	3 (16.7%)
Ceftazidime	-	0	-
Quinolones (Ciprofloxacin or Levofloxacin)	5 (15.6%)	0	3 (16.7%)
Gentamicin	3 (9.4%)	0	2 (8.3%)
Cotrimoxazole	10 (31.3%)	-	2 (11.1%)
Aztreonam	-	4 (16.7%)	-

Abbreviation: GN: Gram negative. “-”: Not tested.

**Table 5 antibiotics-12-00212-t005:** Oral antibiotics prescribed.

Oral Antibiotics Prescribed	Frequency (n)	Percentage (%)
Amocixillin/Clavulanate 500 to 875/125 mg every 8 h	65	30.2
Levofloxacin 500 mg every 12 h	95	44.2
Trimethoprim/sulfamethoxazole 160/800 mg every 12 h	34	15.8
Cloxacillin 500 mg every 6 h	7	3.2
Erythromycin 500 mg every 8 h	6	2.8
Linezolid 600 mg every 12 h	8	3.7
Clindamycin 300 mg every 6	43	20

## Data Availability

Not applicable.

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
