# Peer review of "Bacterial Diversity and Antibiotic Resistance in Patients with Diabetic Foot Osteomyelitis"

_antibiotics, 2023, doi:10.3390/antibiotics12020212_

Round 1

Reviewer 1 Report

Alvaro-Afonso et al report on the diversity of bacteria causing diabetic foot osteomyelitis and antimicrobial resistance (AMR) among them. The fast-evolving nature of the AMR menace requires that AMR traits are routinely investigated among clinical isolates, especially, in invasive infections like osteomyelitis. Thus, the study of Alvaro-Afonso et al is of high clinical and epidemiological significance, particularly, given the limited reports on AMR in bacterial etiologies of osteomyelitis. The manuscript is also well written, except for a few grammatical errors and some minor technical issues/inconsistencies.

Specific comments are as follows:

1.      In the title, the authors should choose whether to retain “susceptibility” or “resistance”, and not both, as the two traits can be inferred from one another.

2.      In Lines 62 to 63, the authors stated that “To the best of our knowledge, no study has de- 62 scribed the specific microbial profile of DFO.”. However, in Lines 233 to 242 of the Discussion, the authors stated categorically that “there are currently few studies on the diversity of microorganisms in patients with DFO” and outlined four studies, with which they contrasted their findings with. The authors need to reconcile this inconsistency between the Introduction and the Discussion.

3.      In Subsection 2.1, the authors need to specify the healthcare facility where the study was conducted.

4.      The content of Subsection 2.6 should be restricted to Statistical Analysis, as captioned, and not include matters pertaining to ethics. Issues regarding ethics should be presented as a separate subsection, 2.7.

5.      Several scientific names are misspelt, such as “Pseudomona aeruginosa” (in Line 58) and “Klebsiella pneumonia”, “Klebsiella oxytora”, and “Citrobacter roseri” (in Table 2). In the said Table 2, there is also a correctly spelt “Klebsiella pneumoniae”, which has the same data as the misspelt “Klebsiella pneumonia”. Also, “Fungus” should be rewritten as “Fungi”, and the incomplete scientific names, namely “Streptococcus”, “Enterococcus”, “Corynebacterium”, and “Serratia” should be completed.

6.      Some of the grammatical issues include the following:

a.       Line 23: The “p” of “p value” should be italicized. This should be done for all other such occurrences.

b.      Line 32: Please rewrite “associated to diabetes” asassociated with diabetes”.

c.       Line 72: Please rewrite “carried between” ascarried out between”.

d. Line 84: Please rewrite “patients whose sample was insufficient…” as “patients whose samples were insufficient…”

Author Response

Dear reviewer,

Thank you very much for your review. Following we are answering concerns that you have detailed in your review. Yellow highlight text indicates the modified or added text in the new version of the manuscript.

  1. In the title, the authors should choose whether to retain “susceptibility” or “resistance”, and not both, as the two traits can be inferred from one another.

We have changed the title, remaining as follows: “Bacterial Diversity and Antibiotic Resistance in Patients with Diabetic Foot Osteomyelitis”.

  1. In Lines 62 to 63, the authors stated that “To the best of our knowledge, no study has de- 62 scribed the specific microbial profile of DFO.”. However, in Lines 233 to 242 of the Discussion, the authors stated categorically that “there are currently few studies on the diversity of microorganisms in patients with DFO” and outlined four studies, with which they contrasted their findings with. The authors need to reconcile this inconsistency between the Introduction and the Discussion.

We have modified this regard in the introduction section:

To the best of our knowledge, few studies have described the specific microbial profile of DFO.

  1. In Subsection 2.1, the authors need to specify the healthcare facility where the study was conducted.

We have included in the subsection 2.1:

A retrospective observational study was carried in the Diabetic Foot Unit of the Complutense University of Madrid…

  1. The content of Subsection 2.6 should be restricted to Statistical Analysis, as captioned, and not include matters pertaining to ethics. Issues regarding ethics should be presented as a separate subsection, 2.7.

We have presented Ethical considerations as separate subsection 2.7.

2.7. Ethical considerations

  1. Several scientific names are misspelt, such as “Pseudomona aeruginosa” (in Line 58) and “Klebsiella pneumonia”, “Klebsiella oxytora”, and “Citrobacter roseri” (in Table 2). In the said Table 2, there is also a correctly spelt “Klebsiella pneumoniae”, which has the same data as the misspelt “Klebsiella pneumonia”. Also, “Fungus” should be rewritten as “Fungi”, and the incomplete scientific names, namely “Streptococcus”, “Enterococcus”, “Corynebacterium”, and “Serratia” should be completed.

Many thanks for your comment. We have corrected the names misspelled. See Table 2:

  1. Some of the grammatical issues include the following:
  1. Line 23: The “p” of “p value” should be italicized. This should be done for all other such occurrences.

We have iyalicized all the “p” of “p value” in the main text.

  1. Line 32: Please rewrite “associated to diabetes” as “associated with diabetes”.

We have rewrited “associated to diabetes” as “associated with diabetes”

  1. Line 72: Please rewrite “carried between” as “carried out between”.

We have rewrited “carried between” as “carried out between”

  1. Line 84: Please rewrite “patients whose sample was insufficient...” as “patients whose samples were insufficient...”

We have rewrited “patients whose sample was insufficient...” as “patients whose samples were insufficient...”

Reviewer 2 Report

Recension of manuscript  No Antibiotics-2150023 „ Bacterial Diversity, Antibiotic Susceptibility, and Resistance in Patients with Diabetic Foot Osteomyelitis, written by Francisco Javier Álvaro-Afonso, Yolanda García-Álvarez , Aroa Tardáguila-García, Marta García-Madrid, Mateo López-Moral, and José Luis Lázaro-Martínez,” which will be published in Antibiotics. 

            The structure of the manuscript is in keeping with the commonly required criteria. The topic of the presented work is very actual. Diabetic foot ulcers are one of the most frequent, serious, and costly complications associated with diabetes mellitus. Diabetic foot infection is related to more severe outcomes contributing to morbidity, increasing costs, and decreasing 

The authors in this study analyzed the bacterial diversity, antibiotic susceptibility, and resistance in patients with complications of diabetic foot osteomyelitis. The authors observed that Proteus spp., Coagulase-negative staphylococci, Staphylococcus aureus, Pseudomonas aeruginosa, Escherichia coli, and Corynebacterium were the most frequently isolated microorganisms and accounted for more than 10% of the diabetic foot osteomyelitis cases. With stratification by Gram-positive (GP) and Gram-negative (GN) bacteria, the authors observed that 91.6% of cultures presented at least one GP bacteria species, and 50.4% gave at least one GN bacteria species. Significant differences were not observed in mean healing time in diabetic foot ulcers with acute osteomyelitis (12.76 weeks (4.50;18)) compared to chronic osteomyelitis (15.31 weeks (7;18.25); p= 0.101) and when comparing cases with soft tissue infection (15.95 (6;20)) and those without such an infection (16.59 (7.25;19.75), p= 0.618). This study shows that when treatment of diabetic foot osteomyelitis is based on early surgical treatment, the type of diabetic foot osteomyelitis and the presence of soft infection are not associated with different or worse prognoses.

Work is clearly legible, brings summarizes new knowledge. The citations are actual, and their format respects usual standards. 

I recommend the manuscript be published. 

Kosice,03. January 2023

Author Response

Dear reviewer, Thank you very much for your comment and review.

Reviewer 3 Report

Dear Authors,

your paper is very interesting. You described bacterial diversity and antibiotic susceptibility in patients with diabetic foot osteomyelitis. You experiments, especially clinically part, were good prepared. Nevertheless, you have to improve part about bacterial resistance. You included in analysis information about inherently resistance of some bacterial, i.e. Pseudomonas aeruginosa.

This bacterium has cephalosporinases ampC-encoded and for this reason in resistant to penicillin and cephalosporins. Even if strain is susceptible in vitro to those antibiotics you can not use them in treatment – there is to high risk of failure of the therapy. Similar is with co-trimoxazole.

I am also wondering about the results for Staphylococcus aureus. You described Staphylococcus aureus and MRSA - do you mean about two different groups of strain: MSSA and MRSA, or among 48 Staphylococcus aureus strains are 16 MRSA strains?

If strains are found as MSSA are susceptible to all cephalosporins, and penicillin with beta-lactamases inhibitors, so I am also wondering about you results – 1 strain resistant to cefuroxime and 2 strain resistant to amoxicillin with clavulanic acid. Resistance to vancomycin in iStaphylococcus aureus is also very rare – check your results.

What recommendation were used to interpret antimicrobial susceptibility testing results? EUCAST or CLSI. You should add to you paper part about it.

Why in table 2 is information about 204 isolates of fungi and number of pathogens is 4 – I do not understand.

What methods were used to examined haemoglobin A1c etc.? Add part about it in you manuscript.

I have also few comments that can help improve your manuscript.

Line 33 – in my opinion you should write “DFI might be related…”

Table 4 – Is Emipenen – should be Imipenem (or another carbapenems)

Line 222 – Number of monomicrobial bone cultures is 59,5%, polymicrobial – 35,5%. Together it is 94,8%. What with the rest to 100%?

Line 294 – should be P. aeruginosa

General – do not use italics for spp.

Author Response

Dear reviewer,

Thank you very much for your review. Following we are answering concerns that you have detailed in your review. Yellow highlight text indicates the modified or added text in the new version of the manuscript.

You included in analysis information about inherently resistance of some bacterial, i.e. Pseudomonas aeruginosa. This bacterium has cephalosporinases ampC-encoded and for this reason in resistant to penicillin and cephalosporins. Even if strain is susceptible in vitro to those antibiotics you can not use them in treatment – there is to high risk of failure of the therapy. Similar is with co-trimoxazole.

Many thanks for your comments. We have reviewed and modified table 4 based on your recommendations.

I am also wondering about the results for Staphylococcus aureus. You described Staphylococcus aureus and MRSA - do you mean about two different groups of strain: MSSA and MRSA, or among 48 Staphylococcus aureus strains are 16 MRSA strains?

We mean about two different groups of strain (48 were MSSA and 16 were MRSA). To clarify it we have modified table 2.

 If strains are found as MSSA are susceptible to all cephalosporins, and penicillin with beta-lactamases inhibitors, so I am also wondering about you results – 1 strain resistant to cefuroxime and 2 strain resistant to amoxicillin with clavulanic acid. Resistance to vancomycin in Staphylococcus aureus is also very rare – check your results.

We have checked our results and we have modified table 3.

What recommendation were used to interpret antimicrobial susceptibility testing results? EUCAST or CLSI. You should add to you paper part about it.

CLSI recommendation were used.  We have included in the method section:

Susceptibility testing was performed in accordance with Clinical and Laboratory Standards by the disk diffusion method.

Reference:

  1. Humphries R, Bobenchik AM, Hindler JA, Schuetz AN. Overview of Changes to the Clinical and Laboratory Standards Institute Performance Standards for Antimicrobial Susceptibility Testing, M100, 31st Edition. J Clin Microbiol. 2021;59(12):e0021321.

Why in table 2 is information about 204 isolates of fungi and number of pathogens is 4 – I do not understand.

It is an error, the number of positive cultures with fungi was 4 and the total number of positive cultures was 204. We have modified in the table 2.

 What methods were used to examined haemoglobin A1c etc.? Add part about it in you manuscript.

Normal values 4.2%-6% were used; standardized according to DCCT/NGSP.  We have included in the results section.

Reference: S M Marshall, J H Barth. Standardization of HbA1c measurements: a consensus statement. Ann Clin Biochem. 2000 Jan;37 (Pt 1):45-6. DOI: 10.1258/0004563001901506.

 I have also few comments that can help improve your manuscript.

Many thanks for your comments:

Line 33 – in my opinion you should write “DFI might be related...”

We have modified in the introduction section.

Table 4 – Is Emipenen – should be Imipenem (or another carbapenems)

We have modified. Is Imipenem

Line 222 – Number of monomicrobial bone cultures is 59,5%, polymicrobial – 35,5%. Together it is 94,8%. What with the rest to 100%?

We have corrected the error, being as follows:

Number of monomicrobial bone cultures is 62.7%, polymicrobial – 37,3%. We have modified these data throughout the main text.

 Line 294 – should be P. aeruginosa General – do not use italics for spp.

Many thanks for all your comments, we have modified P.aeruginosa and we have eliminated italics for spp.

Round 2

Reviewer 3 Report

Dear Authors,

you improved your manuscript, but I found some mistakes in table 3. There is still Staphylococcus aureus as a species (it should be MSSA), I do not agree with results of AST results interpretation for imipenem and CoNS - if CoNS strain is resistant to cloxacillin, it is also resistant to imipenem. Please, check your interpretation. You forgot also about dots in spp. (table 2). Why in table 3 and table 4 in some lines are "-", and in some - 0 - explain it below the tables.

Kind regards

Author Response

Dear reviewer,

Thank you very much for your review. Following we are answering concerns that you have detailed in your review. Yellow highlight text indicates the modified or added text in the new version of the manuscript.

You improved your manuscript, but I found some mistakes in table 3. There is still Staphylococcus aureus as a species (it should be MSSA), I do not agree with results of AST results interpretation for imipenem and CoNS - if CoNS strain is resistant to cloxacillin, it is also resistant to imipenem. Please, check your interpretation. You forgot also about dots in spp. (table 2). Why in table 3 and table 4 in some lines are "-", and in some - 0 - explain it below the tables.

Many thanks for your comments. We have check and modified table 3 based on your suggestions. We have modified table 2, adding dots in spp.  We have explain below the tables 3 and 4 the meaning of “-“.